# Comparison of a Rapid Multiplex Gastrointestinal Panel with Standard Laboratory Testing in the Management of Children with Hematochezia in a Pediatric Emergency Department: Randomized Controlled Trial

Jianling Xie,[a] Kelly Kim,[a] Byron M. Berenger,[b,c] Linda Chui,[d,e] Otto G. Vanderkooi,[f] Silviu Grisaru,[g] Stephen B. Freedman[a,h]

[a]Section of Pediatric Emergency Medicine, Department of Pediatrics, Cumming School of Medicine, University of Calgary, Calgary, Alberta, Canada

[b]Department of Pathology and Laboratory Medicine, Cumming School of Medicine, University of Calgary, Calgary, Alberta, Canada

[c]Alberta Precision Laboratories, Alberta Health Services, Calgary, Alberta, Canada

[d]Department of Laboratory Medicine and Pathology, University of Alberta, Edmonton, Alberta, Canada

[e]Alberta Precision Laboratories, Alberta Health Services, Edmonton, Alberta, Canada

[f]Section of Infectious Diseases, Department of Pediatrics, Cumming School of Medicine, University of Calgary, Calgary, Alberta, Canada

[g]Section of Pediatric Nephrology, Department of Pediatrics, Cumming School of Medicine, University of Calgary, Calgary, Alberta, Canada

[h]Section of Pediatric Gastroenterology, Department of Pediatrics, Cumming School of Medicine, University of Calgary, Calgary, Alberta, Canada

**ABSTRACT** Advances in diagnostic microbiology allow for the rapid identification of a broad range of enteropathogens; such knowledge can inform care and reduce testing. We conducted a randomized, unblinded trial in a tertiary-care pediatric emergency department. Participants had stool (and rectal swabs if stool was not immediately available) tested using routine microbiologic approaches or by use of a device (BioFire FilmArray gastrointestinal panel), which identifies 22 pathogens with a 1-h instrument turnaround time. Participants were 6 months to <18.0 years and had acute bloody diarrhea. Primary outcome was performance of blood tests within 72 h. From 15 June 2018 through 7 May 2022, 60 children were randomized. Patients in the BioFire FilmArray arm had a reduced time to test result (median 3.0 h with interquartile range [IQR] of 3.0 to 4.0 h, versus 42.0 h (IQR 23.5 to 47.3 h); difference of −38.0 h, 95% confidence interval [CI] of −41.0 to −22.0 h). Sixty-five percent (20/31) of participants in the BioFire FilmArray group had a pathogen detected—most frequently enteropathogenic *Escherichia coli* (19%), *Campylobacter* (16%), and *Salmonella* (13%). Blood tests were performed in 52% of children in the BioFire FilmArray group and 62% in the standard-of-care group (difference of −10.5%, 95% CI of −35.4% to 14.5%). There were no between-group differences in the proportions of children administered intravenous fluids, antibiotics, hospitalized, or who had diagnostic imaging performed. Testing with the BioFire FilmArray reduced the time to result availability by 38 h. Although statistical significance was limited by study power, BioFire FilmArray use was not associated with clinically meaningful reductions in health care utilization or improved outcomes.

**IMPORTANCE** Advances in diagnostic microbiology now allow for the faster and more accurate detection of an increasing number of pathogens. We determined, however, that in children with acute bloody diarrhea, these advances did not necessarily translate into improved clinical outcomes. While a greater number of pathogens was identified using a rapid turnaround multiplex stool diagnostic panel, with a reduction in the time to stool test result of over 1.5 days, this did not alter the practice of pediatric emergency medicine physicians, who continued to perform blood tests on a large proportion of children. While our conclusions may be limited by the relatively small sample size, targeted approaches that educate clinicians on the implementation of such technology into clinical care will be needed to optimize usage and maximize benefits.

Address correspondence to Stephen B. Freedman, Stephen.freedman@ahs.ca.

The authors declare a conflict of interest. A BioFire FilmArray Gastrointestinal Panel device was initially provided free of charge by bioMérieux. During the study period, a device was purchased for research purposes by the Alberta Children's Hospital Research Institute. All study test kits were purchased for usage.

**KEYWORDS** rapid multiplex gastrointestinal panel, children, hematochezia, emergency department, randomized controlled trial

Acute bloody diarrhea is a medical emergency for patients of all ages, and infectious etiologies should be considered and prioritized in the evaluation of such patients (1). Among children with acute diarrhea seeking emergency department (ED) care, 11% have bloody diarrhea and 33% of such children have a bacterial pathogen identified (2). Children with bloody diarrhea consume greater resources than those without, specifically, they are more likely to have blood tests performed, to receive intravenous fluids, and experience repeat ED visits (3). In fact, complete blood counts are performed in 46% of children with hematochezia seeking ED care, compared to 16% of those with nonbloody diarrhea (3). Shiga toxin-producing *Escherichia coli* (STEC), in particular *E. coli* O157:H7 and other Shiga toxin 2-producing *E. coli*, are the most concerning bacterial infections to consider in children with acute bloody diarrhea, because of the potential to cause hemolytic uremic syndrome (HUS) (4, 5). As such, the ability to rapidly identify or exclude the presence of these pathogens is important, as detection should trigger frequent monitoring for evidence of evolving microangiopathy (6) and possibly early intravascular volume expansion (7).

The BioFire FilmArray gastrointestinal panel (bioMérieux Canada Inc., Quebec, Canada) is a multiplexed nucleic acid (NA) amplification assay that simultaneously identifies 22 enteric pathogens, including targets for *E. coli* O157:H7 and the Shiga toxin ($stx_1$ and $stx_2$) genes (8). It provides results with a 1-h run time and has excellent diagnostic sensitivity and specificity (9), and its usage has been demonstrated to reduce antibiotic use among hospitalized children with acute diarrhea (10). As such, it has the potential to rapidly identify STEC-infected children at risk for HUS while simultaneously ruling out such infections in the majority of children, thereby reducing the need for close monitoring while awaiting culture results (11).

One of the historical barriers to diagnosing enteric pathogens is the difficulty inherent in obtaining and transporting bulk stool specimen to the diagnostic laboratory for analysis, particularly for outpatients (12). Thus, an important advance in specimen collection includes the use of rectal swabs to expedite specimen acquisition at the point of care. Rectal swab specimens have acceptable sensitivity compared with paired bulk stool specimens (13 to 15), including when tested using the BioFire FilmArray gastrointestinal panel (12), and superior overall yield, due to higher collection rates (15). Thus, we hypothesized that use of the BioFire FilmArray gastrointestinal panel with stool, or rectal swab specimens when the former was not immediately available, would lead to a more targeted monitoring and therapeutic approach to the management of children with acute bloody diarrhea.

To evaluate the impact of the BioFire FilmArray gastrointestinal panel, we conducted a randomized clinical trial to determine if usage in children with hematochezia resulted in a reduction in resource utilization. Secondary objectives included (i) to quantify clinical outcomes among STEC-infected children, (ii) to determine if use of the BioFire FilmArray gastrointestinal panel was associated with greater family satisfaction with care, and (iii) to identify testing criteria to optimize the diagnostic yield of the BioFire FilmArray (i.e., ability to identify bacterial etiologies).

## RESULTS

**Study participants.** From 15 June 2018 through 7 May 2022, 60 children were enrolled and underwent randomization (Fig. 1). Seventeen participants (28%), including 28% (8/29) of those enrolled in the standard-of-care arm, were recruited prior to 1 June 2019, the date that standard-of-care testing was modified. Ninety percent (28/31) of the participants assigned to the BioFire FilmArray arm and 100% (29/29) of those assigned to the standard-of-care arm completed 14-day follow-up. All study participants had their medical records reviewed at day 28 for study outcome data extraction; all were included in the analysis.

Participant median age was 3.7 (interquartile range [IQR], 1.4 to 6.7) years, and 63.3% (38/60) were male. Children allocated to the BioFire FilmArray arm were more likely to

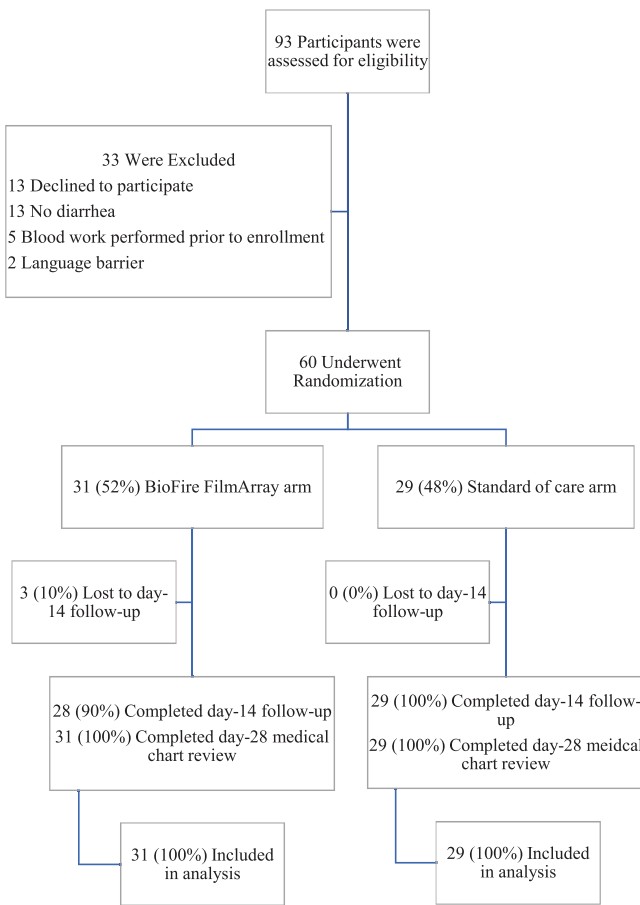

**FIG 1** Enrollment, randomization, and outcomes.

have experienced vomiting (52% versus 14%; $P = 0.003$); otherwise, there were no clinically meaningful differences between groups at the time of enrollment (Table 1).

**Fecal testing results.** Patients in the BioFire FilmArray arm had a shorter median time to stool test result than those in the standard-of-care arm (median [IQR] of 3.0 h [3.0 to 4.0 h] versus 42.0 h (23.5 to 47.3 h); difference of −38.0 h, 95% confidence interval [CI] of [−41.0 to −22.0 h]). Standard-of-care testing was performed on 26 (90%) participants in the standard-of-care arm (2 children submitted specimens that were not suitable for processing by the laboratory and 1 child did not submit a rectal swab or stool specimen) and for 30 (97%) children in the BioFire FilmArray arm (1 child did not submit a rectal swab or stool specimen); those who did not have testing performed never provided a specimen. A bacterial pathogen was identified by standard testing in 37% (11/30) of children in the BioFire FilmArray arm and 35% (9/26) of those in the standard-of-care arm. The pathogens most frequently identified using standard-of-care testing were *Campylobacter* spp. (20%, 11/56) and *Salmonella* spp. (9%, 5/56). Two participants had *E. coli* O157:H7 identified, and one child had a non-O157 STEC detected (Table 2). Standard-of-care *Clostridioides difficile* testing was performed on specimens from 6 study participants and was positive for 1 participant in each study arm; the positive child in the BioFire FilmArray arm was <2 years of age.

All participants in the BioFire FilmArray study arm had a specimen tested on the platform and 20 (65%, with 95% CI of 45 to 81%) had a pathogen identified (Table 2). The most frequently detected bacteria were enteropathogenic *E. coli* (EPEC; 6/31 [19%]), *Campylobacter* (5/31 [16%]), and *Salmonella* (4/31 [13%]) (Table 2). Two participants had a STEC infection in the BioFire FilmArray arm; one of these had *E. coli* O157 ($stx_1$ $stx_2$ positive), while the other was a non-O157 $stx_1$ $stx_2$ positive. Both children received intravenous fluids

**TABLE 1** Baseline characteristics of the enrolled participants

| Characteristic | All participants (N = 60) | Standard-of-care arm (N = 29) | BioFire FilmArray (N = 31) | P value[a] |
|---|---|---|---|---|
| Demographics | | | | |
| Median age, yrs (IQR) | 3.7 (1.4, 6.7) | 3.9 (1.3, 9.8) | 2.7 (1.5, 5.2) | 0.38 |
| Male sex, n (%) | 38/60 (63) | 20/29 (69) | 18/31 (58) | 0.43 |
| | | | | |
| Primary care physician type, n (%) | | | | 0.55 |
| Pediatrician | 14/60 (23) | 8/29 (28) | 6/31 (19) | |
| Family physician | 46/60 (77) | 21/29 (72) | 25/31 (81) | |
| | | | | |
| No. (%) with chronic medical condition(s) | 11/60 (18) | 6/29 (21) | 5/31 (16) | 0.75 |
| Travel outside of Canada or USA in past 4 wks, n (%) | 8/60 (13) | 5/29 (17) | 3/31 (10) | 0.47 |
| Has pets at home, n (%) | 30/60 (50) | 17/29 (59) | 13/31 (42) | 0.30 |
| | | | | |
| Symptoms | | | | |
| Duration (h) of diarrhea, median (IQR) | 54.0 (28.2–122.0) | 54.6 (34.7–132.9) | 50.6 (24.1–121.2) | 0.64 |
| Duration (h) of bloody diarrhea, median (IQR) | 16.3 (12.9–38.1) | 17.0 (12.7–40.5) | 15.2 (12.9–37.5) | 0.55 |
| Maximal no. of diarrheal episodes in any 24-hour period, median (IQR) | 10 (5–14.8) | 10 (5–15) | 10 (5–14) | 0.85 |
| No. of diarrheal episodes in preceding 24 h, median (IQR) | 7 (4–12) | 7 (4–12) | 7 (4–13) | 0.92 |
| Vomiting, n (%) | 20/60 (33) | 4/29 (14) | 16/31 (52) | 0.00 |
| Duration (h) of vomiting, median (IQR) | 0 (0–14.9) | 0 (0) | 0 (0–33.0) | 0.01 |
| Maximal no. of vomiting episodes in any 24-h period, median (IQR) | 0 (0–1) | 0 (0) | 1 (0–1) | 0.01 |
| No. of vomiting episodes in preceding 24 h, median (IQR) | 0 (0–1) | 0 (0) | 0 (0–1) | 0.01 |
| Fever, n (%) | 24/60 (40) | 10/29 (35) | 14/31 (45) | 0.44 |
| Abdominal pain, n (%) | 43/60 (72) | 21/29 (72) | 22/31 (71) | >0.99 |
| | | | | |
| Treatments received prior to enrollment | | | | |
| Prior ED visit for current illness, n (%) | 7/60 (12) | 2/29 (7) | 5/31 (16) | 0.43 |
| Blood test during current illness >12 h preceding triage, n (%) | 2/60 (3) | 1/29 (3) | 1/31 (3) | >0.99 |
| Prior intravenous fluid during current illness, n (%) | 2/60 (3) | 1/29 (3) | 1/31 (3) | >0.99 |
| Prior hospitalization during current illness, n (%) | 0/60 (0) | 0/29 (0) | 0/31 (0) | N/A |
| Took antibiotics in past 60 days, n (%) | 6/60 (10) | 3/29 (10) | 3/31 (10) | >0.99 |

[a]Between-group comparisons used Mann-Whitney U test for continuous variables and chi-square tests and Fisher's exact tests for categorical variables, as appropriate. N/A, not applicable.

in the ED, were discharged home, and were monitored at subsequent health care visits; none developed HUS. *C. difficile* toxin A/B was detected in 7 children on the BioFire FilmArray; however, 5 were < 2 years of age, and thus only 2 were classified as positive for analytic purposes (Table 2). Although no specimens were positive for any parasite targets, 5 (16%) were positive for one of the virus targets.

**Outcomes.** Study groups did not differ in the primary outcome, with 52% (16/31) of children in the FilmArray BioFire group and 62% (18/29) of those in the standard-of-care group having a blood test performed within 72 h of randomization (difference of −10.5%, 95% CI of −35.4 to 14.5%) (Table 3). For the secondary outcomes, there were no between-group differences in the proportions of children (BioFire FilmArray versus standard of care): administered intravenous fluids (35% versus 41%, difference of −6%, 95% CI of −31% to 19%), antibiotics (19% versus 17%, difference of 2.1%, 95% CI −17% to 22%), hospitalized (16% versus 7%, difference or 9%, 95% CI −7% to 25%), or had diagnostic imaging performed (10% versus 7%, difference of 3%, 95% CI −11% to 17%). There were also no differences between study groups in ED length of stay during enrollment visit, the total number of physician visits, and caregiver and patient satisfaction (median score of 9 out of 10 with range of 8 to 10; P = 0.572). No study participants developed HUS, experienced acute kidney injury, or required renal replacement therapy or intensive care unit admission.

The classification and regression tree (CART) model identified duration of bloody diarrhea (≤38 h) and the number of diarrheal episodes in any given 24-h period (≤5 or >9) as being predictive of bacteria identification (Fig. 2). The CART model correctly classified 78.6% (11/14) and 100% (17/17) of those testing negative and positive for bacteria on the BioFire FilmArray panel, respectively. The risk estimate of the CART

**TABLE 2** Specimens submitted and test results[a]

| Study arm and testing performed | All participants (N = 60) | BioFire FilmArray (N = 31) | Standard of care (N = 29) | P value |
|---|---|---|---|---|
| **Standard of care, n (%)** | | | | |
| Organism was detected | 20/56 (36) | 11/30 (37) | 9/26 (35) | >0.99 |
| ≥2 organisms detected | 3/56 (5) | 2/30 (7) | 1/26 (4) | >0.99 |
| Fecal specimens submitted for testing, n (%) | | | | 0.74 |
| No specimen tested | 4 (7) | 1/31 (3) | 3/29 (10) | |
| Stool only | 41 (68) | 23/31 (74) | 18/29 (62) | |
| Rectal swab only | 2 (3) | 1/31 (3) | 1/29 (4) | |
| Rectal swab and stool | 13 (22) | 6/31 (19) | 7/29 (24) | |
| Culture result with rectal swab or stool,[b] n (%) | | | | |
| *Campylobacter* spp. | 11/56 (20) | 6/30 (20) | 5/26 (19) | >0.99 |
| *Salmonella* spp. | 5/56 (9) | 3/30 (10) | 2/26 (8) | >0.99 |
| *E. coli* O157:H7 | 2/56 (4) | 1/30 (3) | 1/26 (4) | >0.99 |
| STEC non-O157 | 1/56 (2) | 1/30 (3) | 0/26 (0) | >0.99 |
| *Shigella* spp. | 0/56 (0) | 0/30 (0) | 0/26 (0) | N/A[c] |
| Shiga toxin positive | 3/3 (100) | 2/2 (100) | 1/1 (100) | N/A |
| *Clostridioides difficile* | 1/6 (17) | 0/5 (20)[d] | 1/1 (100) | 0.17 |
| Viruses, n (%) | | | | |
| Astrovirus | 0/5 (0) | 0/4 (0) | 0/1 (0) | N/A |
| Adenovirus | 0/5 (0) | 0/4 (0) | 0/1 (0) | N/A |
| Norovirus | 0/5 (0) | 0/4 (0) | 0/1 (0) | N/A |
| Rotavirus | 0/7 (0) | 0/5 (0) | 0/2 (0) | N/A |
| Sapovirus | 0/5 (0) | 0/4 (0) | 0/1 (0) | N/A |
| | | | | |
| **BioFire FilmArray, n (%)** | | | | |
| Any positive | N/A | 20 (65) | N/A | N/A |
| >1 positive | N/A | 7 (23) | N/A | N/A |
| Bacteria, n (%) | | | | |
| *Campylobacter* | N/A | 5 (16) | N/A | N/A |
| *Clostridioides difficile* toxin A/B | N/A | 2 (7)[e] | N/A | N/A |
| *Plesiomonas shigelloides* | N/A | 0 (0) | N/A | N/A |
| *Salmonella* | N/A | 4 (13) | N/A | N/A |
| *Vibrio* spp. | N/A | 0 (0) | N/A | N/A |
| *Vibrio cholera* | N/A | 0 (0) | N/A | N/A |
| *Yersinia enterocolitica* | N/A | 1 (3) | N/A | N/A |
| Enteroaggregative *E. coli* | N/A | 1 (3) | N/A | N/A |
| Enteropathogenic *E. coli* | N/A | 6 (19) | N/A | N/A |
| Enterotoxigenic *E. coli* lt/st | N/A | 0 (0) | N/A | N/A |
| Shiga-like toxin-producing *E. coli* Stx1/Stx2 | N/A | 2 (7) | N/A | N/A |
| *E. coli* O157 | N/A | 1 (3) | N/A | N/A |
| *Shigella* or enteroinvasive *E. coli* | N/A | 1 (3) | N/A | N/A |
| Parasites, n (%) | | | | |
| *Cryptosporidium* | N/A | 0 (0) | N/A | N/A |
| *Cyclospora cayetanensis* | N/A | 0 (0) | N/A | N/A |
| *Entamoeba histolytica* | N/A | 0 (0) | N/A | N/A |
| *Giardia lamblia* | N/A | 0 (0) | N/A | N/A |
| Viruses, n (%) | | | | |
| Adenovirus F 40/41 | N/A | 1 (3) | N/A | N/A |
| Astrovirus | N/A | 2 (7) | N/A | N/A |
| Norovirus GI/GII | N/A | 1 (3) | N/A | N/A |
| Rotavirus A | N/A | 0 (0) | N/A | N/A |
| Sapovirus | N/A | 1 (3) | N/A | N/A |

[a]The denominator is 31 for the BioFire FilmArray gastrointestinal panel group and 29 for the standard of care group, unless otherwise specified.
[b]Either sample type was positive. Testing was modified with culture discontinued and replaced by testing using the BDMax in June 2019.
[c]N/A, not applicable.
[d]One child <2 years of age was positive for *Clostridioides difficile* based on detection of the toxin gene by PCR and was not included in the number reported as positive, since this likely represented colonization.
[e]Five children <2 years of age were *Clostridioides difficile* toxin A/B positive on the BioFire test; these children were not included in the number reported as positive, since this likely represented colonization.

**TABLE 3** Study outcomes[a]

| Outcome | All participants (N = 60) | BioFire FilmArray (N = 31) | Standard of care (N = 29) | P value[b] | Adj P[b] | Difference (95% CI) |
|---|---|---|---|---|---|---|
| Primary outcomes | | | | | | |
| Blood test performed within 72 h of randomization, n (%) | 35/60 (58) | 17/31 (55) | 18/29 (62) | 0.57 | >0.99 | −7.2% (−32.1 to 17.6%) |
| Any blood test performed, n (%) | 37/60 (62) | 18/31 (58) | 19/29 (66) | 0.55 | >0.99 | −7.5% (−32.0 to 17.1%) |
| Secondary outcomes | | | | | | |
| Intravenous fluid administered, n (%) | 23/60 (38) | 11/31 (36) | 12/29 (41) | 0.79 | >0.99 | −5.9% (−30.5 to 18.7%) |
| No. of physician visits (ED and non-ED), median (IQR) | 0 (0–1) | 0 (0–1) | 1 (0–1) | 0.80 | >0.99 | 0 (0–0) |
| ED length of stay (h) during index visit, median (IQR)[c] | 4.6 (3.4–5.6) | 4.6 (3.6–5.7) | 4.2 (2.9–5.6) | 0.55 | >0.99 | 0.28 (−0.72 to 1.20) |
| Antibiotics administered, n (%) | 11/60 (18) | 6/31 (19) | 5/29 (17) | >0.99 | >0.99 | 2.1% (−17.4 to 21.7%) |
| Hospitalization, n (%) | 7/60 (12) | 5/31 (16) | 2/29 (7) | 0.43 | >0.99 | 9.2% (−6.7 to 25.1%) |
| Diagnostic imaging performed, n (%) | 6/60 (10) | 4/31 (13) | 2/29 (7) | 0.67 | >0.99 | 6.0% (−9.0 to 21.0%) |
| Caregiver and patient satisfaction related to ED visit, median (IQR)[d] | 9 (8–10) | 9 (8–10) | 9 (8–10) | 0.57 | >0.99 | 0 (0–1) |
| Time interval (h) from triage to stool test lab result, median (IQR) | 7.0 (3.0–34.5) | 3.0 (3.0–4.0) | 42.0 (23.5–47.3) | <0.00 | <0.001 | −38.0 (−41.0 to −22.0) |

[a]Data include all events from the index ED visit until day 14 of follow-up, unless otherwise specified. No children experienced the outcomes of intensive care unit admission, hemolytic uremic syndrome, acute kidney injury, or renal replacement therapy.
[b]Between-group comparisons used the Mann-Whitney U test for continuous variables and chi-square tests and Fisher's exact tests for categorical variables, as appropriate. The Holm method was used to adjust P values for multiple comparisons.
[c]Included only those who were discharged from the emergency department (N = 27 in each study arm).
[d]Reported on a scale of 0 to 10. Data were missing for 3 children in the BioFire FilmArray arm.

model was 0.097 (standard error, 0.053) in a resubstitution analysis and 0.387 (standard error, 0.087) in cross-validation analysis.

## DISCUSSION

In this study, children whose specimens were tested with the BioFire FilmArray had a much shorter time to test result. However, testing stool specimens provided by children who presented to the ED with diarrhea and hematochezia with the BioFire FilmArray was not associated with reductions in health resource utilization. Also, testing specimens with the BioFire FilmArray was not associated with greater patient satisfaction or improved clinical outcome.

Previous observational studies in adults reported that the BioFire FilmArray gastrointestinal panel, compared to conventional stool testing, accurately identified pathogens while reducing turnaround times. This facilitated a reduction in the duration of isolation and ED length of stay, while also reducing inappropriate antibiotic use and diagnostic imaging (16–20). A study of children with diarrhea also reported a reduction in antibiotic administration with use of the BioFire FilmArray (10). Our findings do not align with those of previous reports, but there are several potential explanations.

Previous studies included patients with general diarrheal disease while we only included patients with hematochezia. These populations have inherent differences: children with hematochezia, compared to those without, are more likely to have a bacterial pathogen identified (33.0% versus 7.9%, difference of 25.1, 95% CI of 16.3 to 33.9%), less likely to have viruses detected (31.3% versus 72.3%, difference of −41.0%, 95% CI of −49.9 to −32.1%), and are more likely receive antibiotics (3). Thus, such patients are viewed as having a more severe illness and clinicians are therefore less likely to modify their treatment approach.

The greater number of children in the BioFire FilmArray arm who experienced vomiting (52%) compared with those in the standard-of-care arm (14%) may have reduced the potential benefit we hypothesized would be seen. This proposition is based on knowledge children with isolated diarrhea (i.e., diarrhea in the absence of vomiting) are less likely to receive intravenous fluids compared to children with vomiting (21).

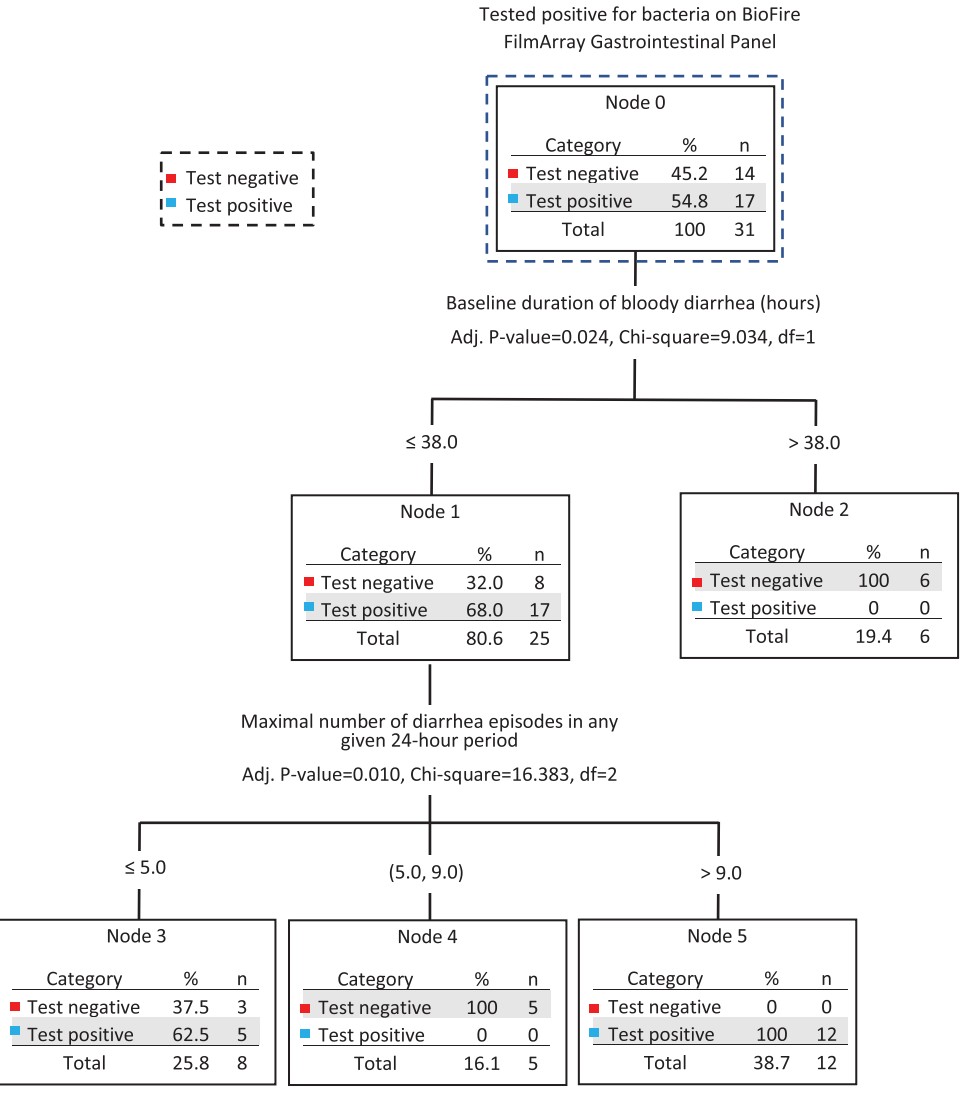

**FIG 2** Classification and regression tree analysis for predicting bacteria identification with the BioFire FilmArray gastrointestinal panel. The analysis identified baseline duration of bloody diarrhea (≤38 h) and the number of diarrhea episodes in any given 24-h period (≤5 or >9 episodes) as prediction of testing positive for bacteria target. Testing positive for bacteria was not observed in study participants without these clinical findings.

It is possible that as guidelines for managing infectious diarrhea evolve, and physician adherence improves (22–26), practice patterns for managing diarrhea and hematochezia may become less variable. For example, the recent guideline from the Infectious Disease Society of America suggested oral rehydration as the first-line therapy of rehydration, and empirical antibiotics is not recommended for bloody diarrhea or for most acute watery diarrheal illness (27). Although these two treatment decisions are independent, the differences between the two study arms in our study were small and not clinically significant. As this study was conducted in a pediatric tertiary-care ED, guideline adherence may have been greater than occurs in nonpediatric tertiary-care centers. For example, low-value radiograph use is more commonly requested by emergency medicine physicians practicing in EDs that primarily treat adult patients (28).

A greater number of bacteria were detected in paired specimens collected from children in the BioFire FilmArray group when tested using the BioFire FilmArray ($n = 23$) compared to standard-of-care testing ($n = 11$). However, we did not find this added knowledge to have significant clinical implications. In keeping with previous reports (29, 30), the most common bacterial target identified by the BioFire FilmArray

was EPEC. In most cases, antimicrobial treatment of such infections is unnecessary (31, 32), and EPEC is often codetected along with other common enteropathogens, such as norovirus, *Salmonella*, and *Campylobacter* (30).

Although use of the BioFire FilmArray gastrointestinal panel in hospitalized adults, at a cost of US$155 per test, has been shown to be cost-effective through reductions in ancillary diagnostic tests, unnecessary antibiotics, and shorter lengths of stay (19), a randomized trial in Canada found no differences in hospital resource utilization compared with a conventional stool testing group (33). In the latter study, the estimated cost of the BioFire FilmArray gastrointestinal panel was Can$180, which is approximately 40% more expensive than conventional testing methods (33). These conflicting findings were summarized in a United Kingdom-based systematic review and economic analysis that identified significant uncertainties and thus need for future research to further elucidate the economic and clinical implications of the use of such technology (34).

**Limitations.** The study was conducted in children with diarrhea and hematochezia who presented to ED for care; thus, the results may not be generalizable to other populations and settings. Although our study has the strength of being a randomized trial, there may be response bias from the patients and physicians due to lack of blinding of the study arm allocation. However, treatments such as intravenous fluids and antibiotics administration usually depend on patient clinical status and the enteropathogen identified; thus, the impact on the study outcome due to nonblinding of allocation likely is minimal.

Our lack of effect may reflect the large difference we sought to find, the fact that many physicians routinely perform blood tests on children with bloody diarrhea, that physicians may not be willing to wait for the BioFire FilmArray test result before making such a decision, and a lack of physician familiarity with the platform and how to optimally integrate results into clinical care. In addition, the transition in the standard-of-care arm from culture to a molecular platform may have impacted our results; however, the latter was limited to a four-pathogen panel and the overall difference in test turnaround time remained significant. We also lacked power to detect many of the secondary outcomes, we did not document the effect on isolation practices, and we had very few STEC-infected children, which was a key driver behind the study's rationale. As smaller differences may be clinically meaningful and further exposure to rapid molecular platforms may alter physician practice patterns, further research exploring the benefits of rapid enteric diagnostic platforms is required.

**Conclusion.** Although testing with the BioFire FilmArray gastrointestinal panel reduced the time to result availability by over 50 h and identified a greater number of pathogens, use was not associated with a reduction in health care utilization, improved outcomes, or greater patient or parent satisfaction. However, as this was a small, single-center study, larger, multicenter studies are needed. Moreover, in the context of certain infections such as STEC, the reduced turnaround time has the potential to significantly alter clinical care.

## MATERIALS AND METHODS

**Trial design.** In this single-center, randomized, unblinded trial, participants with acute bloody diarrhea were enrolled in the Alberta Children's Hospital pediatric ED located in Calgary, Alberta, Canada. Participants were randomized to have stool testing performed using the BioFire FilmArray gastrointestinal panel or routine testing for enteric bacteria. Rectal swabs were performed in the ED if stool specimens were unavailable. A stool specimen was requested of all children who provided a rectal swab initially, as per routine specimen collection procedures. Home stool collection was performed by those unable to provide a sample while in the ED.

**(i) Ethics statement.** Parents or guardians provided written informed consent for their children to participate; assent was obtained from the child when appropriate. The University of Calgary Research ethics board approved the study, which was registered at ClinicalTrials.gov (identifier NCT03362970).

**Trial participants.** We prospectively identified children who presented with hematochezia to the ED. Children were eligible if they were aged 6 months to 17.99 years of age and had ≥3 episodes of diarrhea within the preceding 24 h, with at least one of these episodes having blood identified in the stool. Children with any of the following were excluded: (i) previously enrolled in the study; (ii) unavailable for day-14 follow-up; (iii) known to be neutropenic (neutrophil count of <1,000/dL), or at high risk of being

neutropenic (receiving chemotherapy); (iv) blood work performed within preceding 12 h; (v) documented evidence of a STEC infection based on a specimen submitted for testing during the current illness; (vi) preexisting diagnosis of inflammatory bowel disease; or (vii) language barrier that prevented the ability to obtain informed consent and assent.

**Randomization.** Random-number-generating software, accessed through the Web-based REDCap randomization tool, with a 1:1 trial-group assignment ratio was used to sequentially assign children to the BioFire FilmArray gastrointestinal panel or standard testing. As the results of the testing were provided to the responsible physician as soon as they were available to permit an assessment of the impact of the intervention on clinical care, after allocation was performed blinding was not feasible.

**Procedures.** In both study groups, once consent was obtained and randomization performed, the responsible physician was informed of the result of the allocation. Subsequently, baseline clinical data were collected. When performed, rectal swabs were done using the Copan FecalSwab (Copan Italia, Bresica, Italy). Following the completion of the visit, data regarding all testing, procedures, and medications administered was manually collected by the research assistant from the medical record. The family was contacted 14 days later to collect outcome information (i.e., clinical and health resource use) and satisfaction (reported on a 0-to-10 scale). Medical records and select administrative databases, which include all EDs within the Calgary region, were reviewed to ascertain and confirm outcome data on day 28.

**(i) Standard of care.** If stool was unavailable, rectal swab testing was performed. The specimens, once available, were submitted for routine enteric bacterial culture at a College of Physician and Surgeons of Alberta-accredited laboratory (Alberta Precision Laboratories, Calgary, Alberta, Canada) as per laboratory protocols. Culture was performed for *Aeromonas*, *Campylobacter*, *E. coli* O157:H7, *Salmonella*, *Shigella*, and *Yersinia*. For rectal swabs, tubes were vortexed and 100 $\mu$L of the modified FecalSwab medium was plated and streaked for isolation. For enrichment broths, 200 $\mu$g of solid stool or around 200 $\mu$L of liquid stool was used. All stool specimens and rectal swabs were plated on the following agars supplied by Dalynn Biologicals (Calgary, Alberta, Canada) unless otherwise specified: sheep blood agar, MacConkey agar with crystal violet, Hektoen agar or CHROMagar *Salmonella* (CHROMAgar, Paris, France), CHROMagar STEC, cefsulodin-igrasan novobiocin agar (CIN), and *Campylobacter* blood-free agar. *Campylobacter* plates were incubated under microaerophilic conditions at 42°C; all other media were incubated at 35°C $\pm$ 2°C.

On 1 June 2019, the clinical laboratory began testing all specimens using the BDMax enteric bacterial panel (Becton, Dickinson, Mississauga, Ontario, Canada) which is a PCR test for *Salmonella* spp., *Shigella* spp. and enteroinvasive *E. coli*, *Campylobacter* spp. (*C. jejuni* and *C. coli*), and STEC (*stx1* and *stx 2*). The panel requires 3 h of run time and was batch run by the clinical laboratory based on the volume of tests requested. Stool samples were also planted to a blood agar and CIN plate. All NA-positive specimens subsequently underwent culture as described above with the addition of a Gram-negative broth for *Shigella*, *Salmonella*, and STEC-positive specimens planted to the appropriate agar if the direct stool was negative. Prior to June 2019, all positive STECs were based on culture on STEC CHROMAgar with supplemental Shiga toxin detection performed using the Shiga toxin QuikChek (TechLab, Inc., Blacksburg, VA). After that time, Shiga toxin detection was based on PCR using the BDMax followed by culture.

Testing for viruses, parasites, and *C. difficile* was also performed if requested by the responsible physician. Conventional virus testing was performed using an in-house, real-time, quantitative PCR virus panel that detected generic adenovirus, astrovirus, norovirus genogroups I and II, rotavirus, and sapovirus (35). Conventional parasite screening was performed using the BDMax enteric parasite panel for *Giardia*, *Cryptosporidium*, and *Entamoeba histolytica*. Microscopy was performed if requested and additional history was provided to the laboratory (36). *C. difficile* testing was performed using the Liaison *C. difficile* glutamate dehydrogenase (GDH) enzyme immunoassay (EIA; DiaSorin S.p.A.; Italy) to screen specimens. If the GDH EIA was positive, the specimen was tested by PCR for the presence of toxin B on the Xpert *C. difficile* test (Cepheid, Sunnyvale, CA). Invalid specimens on the Xpert *C. difficile* were retested using the *C. diff* QuikChek Complete EIA (Techlab, Inc.).

**(ii) BioFire FilmArray.** If immediately available, stool, and if stool was unavailable a rectal swab, was collected by the study research assistant who tested the specimen on the BioFire FilmArray gastrointestinal panel. The latter contains all necessary reagents for sample preparation, NA extraction, and purification, which is performed from the unprocessed specimens and automatically by the system whose software analyzes the endpoint melting curve for each target on the panel. The array can detect 22 analytes: *Campylobacter* (*C. jejuni*, *C. coli, and C. upsaliensis*), *Salmonella*, *Shigella* enteroinvasive *E. coli*, *Vibrio* (*V. parahaemolyticus*, *V. vulnificus*, and *V. cholerae*), *Yersinia enterocolitica*, Shiga-like toxin-producing *E. coli* (STEC) $stx_1$ $stx_2$, enterotoxigenic *E. coli* (ETEC) *lt st E. coli* O157, *C. difficile* (toxin A/B), *Plesiomonas shigelloides*, enteroaggregative *E. coli*, EPEC, norovirus GI/GII, rotavirus A, adenovirus F40/41, astrovirus, sapovirus (I, II, IV, and V), *Cryptosporidium*, *Entamoeba histolytica*, *Giardia lamblia*, and *Cyclospora cayetanensis*. After mixing, a 200-$\mu$L aliquot of stool suspension in Cary-Blair transport medium was used for the BioFire FilmArray test, which was conducted according to the manufacturer's instructions.

Prior to study launch, the ED physicians were educated about the expected turnaround time of the BioFire FilmArray, the results that it would provide, and to consider delaying other investigations, if clinically appropriate, while awaiting results. The result of the BioFire FilmArray test was printed and brought to the ED, where it was provided to the ED physician for review. Treatment decisions were at the sole discretion of the ED treating physician. In addition to testing using the BioFire FilmArray, participants had standard-of-care testing as described above once a suitable specimen was available.

**Outcomes.** The primary outcome was the performance of any blood test within 72 h following randomization. This outcome was selected as it was hypothesized that rapid pathogen identification would resolve diagnostic uncertainty, leading to a reduction in blood testing. This is an important goal, as two-thirds of children undergoing venipuncture are very anxious about the procedure, which

causes moderate to severe pain in a similar proportion (37). Moreover, these early painful experiences are associated with long-term health consequences, including needle phobia and health care avoidance (38, 39).

Secondary outcomes specified *a priori* included the following: (i) intravenous fluid administration (index visit through day 14 of follow-up); (ii) total subsequent physician visits (ED and non-ED index visit through day 14 of follow-up); (iii) ED length of stay during the index visit; (iv) antibiotic administration; (v) hospital and intensive care unit admission; (vi) diagnostic imaging performed; (vii) development of HUS; (viii) acute kidney injury, categorized based on chart review in accordance with KDIGO guidelines (40); (ix) use of renal replacement therapy; and (x) caregiver and patient satisfaction with the care received during the ED visit on a 10-point Likert scale, as reported on day 14 of follow-up. Finally, we sought to identify clinical features that could be used to optimally identify bacterial enteropathogens with the BioFire FilmArray.

**Sample size.** We assumed that the blood testing would be performed in 60% of the children in the standard-of-care group within 72 h of enrollment. This estimate was based on prior research that 46% of children with hematochezia had blood work performed at the index ED visit, and an additional 12% revisited the ED and had blood work subsequently performed (3). At a significance level of 5%, a sample size of 54 participants would provide the trial with 80% power to detect an absolute between-group difference of 40% in the outcome. This reduction corresponded to blood testing being performed in 20% of children with hematochezia, which was deemed by our team to reflect the proportion for whom blood work would be truly indicated in children for whom knowledge of the infectious etiology had been elucidated. We intended to recruit 60 participants to allow for a rate of loss to follow-up of 10%. All tests of hypotheses were two-sided. No data and safety monitoring committee interim analyses were planned.

**Statistical analysis.** Data were summarized with counts and percentages for categorical variables and with medians and interquartile ranges for continuous variables. The outcome variables were compared between the randomization groups, using chi-square and Fisher's exact tests for categorical variables and Mann-Whitney U test for continuous variables.

Because the detection of *Clostridioides difficile* in children <2 years of age is usually not clinically significant (41, 42), detection in such children was classified as a negative test result in the analyses. Time to specimen result was calculated as the time from ED triage until result provided. The specimen with the shortest time interval was considered (i.e., if a swab specimen was collected and result was available prior to a stool specimen, the former time interval was employed for analytic and reporting purposes).

To identify clinical features that could guide detection of bacteria with the BioFire FilmArray, we used the CART method to construct a decision tree by recursively partitioning feature space for all *a priori*-identified candidate predictor variables into groups, following the approach of impurity reduction, until no further division was possible. The first division was selected as the single best classification for the outcome variable. Variables evaluated for inclusion in the model included the following: age, sex, access to a primary care physician, type of primary care physician (e.g., pediatrician or family physician), presence of a chronic condition, fever and abdominal pain, antibiotic use in past 60 days, travel abroad in past 4 weeks, having pets at home, diarrhea and vomiting duration and maximal number of episodes in any 24-h period and in the 24 h prior to the ED visit, duration of bloody diarrhea, presence of vomiting, a prior ED visit, blood test, intravenous rehydration, and hospitalization for the current illness. A 10-fold cross-validation was performed to assess the tree structure's generalizability to a larger population.

All analyses were specified *a priori*. We included data from all participants who underwent randomization in accordance with the intention-to-treat principle. Multiple imputation was not required due to the low frequency of missing data. The overall significance level for statistical tests of secondary and outcomes was set at 0.05 with the intention to use the Holm method to adjust for multiple comparisons if any were significant (43). Analyses were performed with SPSS software, version 24.0.0.1 (IBM).

**Data availability.** Study data will be made available upon request to the corresponding author to individuals with appropriate ethics approvals and established data-sharing agreements.

## ACKNOWLEDGMENTS

We acknowledge the assistance and support of Gillian Currie, Daniel Gregson, Brent Hagel, and Susan Samuel in study design.

Financial support for the conduct of the study was provided by the Alberta Children's Hospital, Department of Pediatrics Innovation Award. Stephen B. Freedman is also supported by the Alberta Children's Hospital Foundation Professorship in Child Health and Wellness. A BioFire FilmArray gastrointestinal panel device was initially provided free of charge by bioMérieux. During the study period, a device was purchased for research purposes by the Alberta Children's Hospital Research Institute. All study test kits were purchased for usage. All other tests were part of clinical care and were included in the services provided to Alberta residents as part of the provincial health insurance plan. None of the funders (including bioMérieux) had any input into the design or conduct of the trial; the collection, management, analysis, or interpretation of the data; the preparation, review, or approval of the manuscript; or the decision to submit the manuscript for publication.

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
