## [Reviewer comments · Microbiology Spectrum]

Microbiology Spectrum

Comparison of a Rapid Multiplex Gastrointestinal Panel with Standard Laboratory Testing in the Management of Children with Hematochezia in a Pediatric Emergency Department: Randomized Controlled Trial

Jianling Xie, Kelly Kim, Byron Berenger, Linda Chui, Otto Vanderkooi, Silviu Grisar, and Stephen Freedman

Corresponding Author(s): Stephen Freedman, University of Calgary

Review Timeline:

Submission Date:	January 18, 2023
Editorial Decision:	February 28, 2023
Revision Received:	March 8, 2023
Accepted:	March 10, 2023

Editor: Paul Luethy

Reviewer(s): Disclosure of reviewer identity is with reference to reviewer comments included in decision letter(s). The following individuals involved in review of your submission have agreed to reveal their identity: James J Dunn (Reviewer #2)

Transaction Report:

DOI: <https://doi.org/10.1128/spectrum.00268-23>

February 28, 2023

Dr. Stephen Freedman
University of Calgary
Pediatrics
28 Oki Drive NW
Calgary, Alberta T3B 6A8
Canada

Re: Spectrum00268-23 (Comparison of a Rapid Multiplex Gastrointestinal Panel with Standard Laboratory Testing in the Management of Children with Hematochezia in a Pediatric Emergency Department: Randomized Controlled Trial)

Dear Dr. Stephen Freedman:

Link Not Available

Sincerely,

Paul Luethy

Journals Department
Reviewer comments:

Reviewer #1 (Comments for the Author):

This is an interesting study that attempts to address important considerations relevant to diagnostic stewardship in a pediatric population. The sample size is small, and the trial was not blinded, but both are reasonable given the nature of this single-center study and the two treatments being compared. The significance of this paper is that it lays the groundwork for larger studies.

This manuscript is well-written, but I suggest proofreading for minor grammatical and spelling errors throughout. Additionally, I would remove the word "the" before "HUS" throughout the manuscript.

Study design: were the ED clinicians provided any guidance to withhold blood testing or other interventions until after stool samples/rectal swabs were sent? Time to these interventions would have been interesting to measure in addition to the binary outcome of whether each intervention was performed since, as noted in the discussion, clinical status might have been the primary driver of the decision to obtain blood tests or administer therapy.

Lines 265-266: Suggest rephrasing this to indicate that the detection of *C. difficile* in children <2 years of age is usually clinically insignificant rather than emphasizing that *C. difficile* is found in asymptomatic children since the population studied here is inherently symptomatic.

Line 324: Please clarify what is meant by *C. difficile* testing. One could argue that all patients in the BioFire arm had *C. difficile* testing. This sentence also makes the subsequent paragraph difficult to understand; how many children <2 years of age were positive for *C. difficile* on the BioFire?

Discussion: The authors very appropriately identify that clinicians' approach to the treatment of children with hematochezia is unlikely to be altered by the rapid identification of a pathogen and the potential influence of guidelines in the standardization of care.

- I would, however, separate decisions to use antibiotics from decisions to obtain blood tests and administer fluids.
- I noted that there was no discussion of the costs of the BioFire vs. standard of care. Although this is not a cost-effectiveness analysis, it might be worth mentioning the financial tradeoff of one approach vs. the other.
- Lines 401-403: please check for syntax and grammatical errors

Conclusion: I agree with the modest conclusions drawn and would stipulate that this was a small, single-center study and suggest that larger, multi-center studies are needed.

Reviewer #2 (Comments for the Author):

The study by Xie et al. details a randomized controlled trial of the utility of a multiplex, molecular gastrointestinal panel in pediatric patients presenting to the ED with hematochezia compared to standard of care (SOC) testing practices. The primary outcome of blood testing within 72 hours of presentation was not significantly different between the GI panel and SOC groups. Nor were other secondary outcomes such as IV fluid administration, hospitalization, and antibiotic administration. The authors also developed a CART model which predicted whether a bacterial target would be detected based on duration of bloody diarrhea and number of diarrheal episodes in a 24-hour period.

The study was well-designed and the paper clearly and concisely written. The only caveat to the study design was the change in SOC testing methods to include the BD MAX enteric bacterial panel PCR in the third year of the trial. Approximately 80% of the participants in the SOC group had been enrolled by that time.

Line 59: Should also include rectal swab specimens here in addition to stool.

Line 141: Indicate here that patients were randomized to have stool and/or rectal swab testing performed using the BioFire panel or conventional testing. The information regarding home stool collection could also be moved here (lines 193-196).

Line 156: Further define what is meant by "known to be STEC positive". Does this mean a patient with a recent previous visit in which STEC was identified? Please clarify.

Line 179: Should say "rectal swab testing was performed" or "a rectal swab was obtained".

Line 205: Further details of the assays or modalities for virus and parasite testing are needed. From Table 2 it appears that six patients had *C. difficile* toxin testing performed as standard of care but details of that assay do not appear in the Methods.

Line 220: If rectal swabs were tested using the BioFire panel then details of test preparation with that specimen type should be included.

Table 2: By ">1 positive" do the authors mean to indicate the number of patients in which more than one organism was detected? In the SOC testing, does shiga toxin positive refer to the PCR assay? It did not appear that an EIA was used for shiga toxin detection. Please clarify. In the footnotes indicating children <2 years of age with a positive *C. difficile* toxin, the authors should add that the numbers were not included since this likely represents colonization. Additional details are needed for reference 2.

Staff Comments:

Preparing Revision Guidelines

Please return the manuscript within 60 days; if you cannot complete the modification within this time period, please contact me. If you do not wish to modify the manuscript and prefer to submit it to another journal, please notify me of your decision immediately so that the manuscript may be formally withdrawn from consideration by Microbiology Spectrum.

**DEPARTMENT OF PEDIATRICS
ALBERTA CHILDREN'S HOSPITAL
28 Oki Drive NW
Calgary, AB Canada T3B 6A8**

February 28, 2023

Manuscript #: Spectrum00268-23

Title: Comparison of a Rapid Multiplex Gastrointestinal Panel with Standard Laboratory Testing in the Management of Children with Hematochezia in a Pediatric Emergency Department: Randomized Controlled Trial

Dear Dr. Luethy,

Thank you for providing us the opportunity to address the comments provided by your Reviewers. We have carefully considered all comments and requirements and have revised our manuscript accordingly whenever possible. We address each comment on a point-by-point basis below. We hope that our responses are acceptable and we would be happy to discuss any outstanding concerns.

Thank you for reviewing our manuscript.

Sincerely,

Stephen Freedman, MDCM, MSc, FAAP, FRCPC
Sections of Paediatric Emergency Medicine and Gastroenterology
Alberta Children's Hospital
Alberta Children's Hospital Foundation, Professor in Child Health and Wellness
University of Calgary

Comment	Response	Changes/Location
Reviewer #1		
This manuscript is well-written, but I suggest proofreading for minor grammatical and spelling errors throughout. Additionally, I would remove the word "the" before "HUS" throughout the manuscript.	Thank you for pointing out these errors which we have addressed in this revision.	
Study design: were the ED clinicians provided any guidance to withhold blood testing or other interventions until after stool samples/rectal swabs were sent? Time to these interventions would have been interesting to measure in addition to the binary outcome of whether each intervention was performed since, as noted in the discussion, clinical status might have been the primary driver of the decision to obtain blood tests or administer therapy.	Indeed, the ED MDs were educated about the BioFire, the anticipated turnaround time and the information that it would convey. We explained that the information could inform the diagnosis and management and that if not clinically indicated, they could potentially avoid other investigations until they receive the BioFire result. This information has been added to the manuscript. Unfortunately, we did not collect the time of laboratory testing or other interventions and thus do not know timing in relation to receipt of the BioFire result.	Page 12: “Prior to study launch, the ED physicians were educated about the expected turnaround time of the BioFire FilmArray, the results that it would provide, and to consider delaying other investigations, if clinically appropriate, while awaiting results.”
Lines 265-266: Suggest rephrasing this to indicate that the detection of C. difficile in children <2 years of age is usually clinically insignificant rather than emphasizing that C. difficile is found in asymptomatic children since the population studied here is inherently symptomatic.	Thank you for pointing out this distinction and we have revised the sentence accordingly.	Page 13: “Because the detection of Clostridioides difficile in children < 2 years of age is usually not clinically significant,^{22, 23} detection in such children was classified as a negative test result in the analyses.”
Line 324: Please clarify what is meant by C. difficile testing. One could argue that	We apologize for the confusion and have clarified that was is referred to here is	Page 11: “Testing for viral, parasites, and Clostridioides difficile was also performed if

Comment	Response	Changes/Location
all patients in the BioFire arm had C. difficile testing. This sentence also makes the subsequent paragraph difficult to understand; how many children <2 years of age were positive for C. difficile on the BioFire?	standard of care C. difficile testing [Liaison C. difficile glutamate dehydrogenase (GDH) enzyme immunoassay (EIA; DiaSorin S.p.A.; Italy); if GDH EIA was positive the specimen was tested by polymerase chain reaction for the presence of toxin B on the Xpert® C. difficile test (Cepheid; Sunnyvale, CA). Invalid specimens on the Xpert® C. difficile were retested using the C. diff Quik Chek Complete® EIA (TECHLAB, Inc; Blacksburg, VA)].	requested by the responsible physician. Conventional virus testing was performed using an in-house, real-time, quantitative polymerase chain reaction virus panel that detects generic adenovirus, astrovirus, norovirus genogroups I and II, rotavirus, and sapovirus.¹⁶ Conventional parasite screening was performed using the BDmax enteric parasite panel for Giardia, Cryptosporidium, and Entamoeba histolytica. Microscopy was performed if requested and additional history was provided to the laboratory.¹⁷ C. difficile testing was performed using the Liaison C. difficile glutamate dehydrogenase (GDH) enzyme immunoassay (EIA; DiaSorin S.p.A.; Italy) to screen specimens. If the GDH EIA was positive the specimen was tested by polymerase chain reaction for the presence of toxin B on the Xpert® C. difficile test (Cepheid; Sunnyvale, CA). Invalid specimens on the Xpert® C. difficile were retested using the C. diff Quik Chek Complete® EIA (TECHLAB, Inc; Blacksburg, VA)].” Page 17: “Standard of care C. difficile testing was performed on specimens from 6 study participants and was positive for one participant in each study arm; the positive child in BioFire FilmArray arm was <2 years of age.”

Comment	Response	Changes/Location
	In addition, all children enrolled in the BioFire arm had C. difficile testing performed and 7 were positive however 5 were < 2 years of age and thus we report those as negative in the table.	Page 17: “C. difficile toxin A/B was detected in 7 children on the BioFire FilmArray, however five were < 2 years of age, thus only 2 are classified as positive for analytic purposes; Table 2.”
I would, however, separate decisions to use antibiotics from decisions to obtain blood tests and administer fluids.	We have identified more clearly that these two treatment decisions are independent of one another.	Page 20: “Although these two treatment decisions are independent, the differences between the two study arms in our study were small and not clinically significant. As this study was conducted in a pediatric tertiary care ED, guideline adherence may have been greater than occurs in non-pediatric tertiary care centers.”
I noted that there was no discussion of the costs of the BioFire vs. standard of care. Although this is not a cost-effectiveness analysis, it might be worth mentioning the financial trade-off of one approach vs. the other.	We have added a paragraph to address these important economic considerations.	Page 21: “Although use of the BioFire FilmArray Gastrointestinal Panel in hospitalized adults, at a cost of \$155 USD/test, has been shown to be cost effective through reductions in ancillary diagnostic tests, unnecessary antibiotics and shorter lengths of stay,²⁸ a randomized trial in Canada found no differences in hospital resource utilization compared with a conventional stool testing group.⁴² In the latter study, the estimated cost of the BioFire FilmArray Gastrointestinal Panel was \$180 CDN, which is approximately 40% more expensive than conventional testing methods.⁴² These conflicting findings are summarized in a UK-based systematic review and economic analysis that identified

Comment	Response	Changes/Location
		significant uncertainties and thus need for future research to further elucidate the economic and clinical implications of the use of such technology. ⁴³ ”
Lines 401-403: please check for syntax and grammatical errors	We have revised this sentence which we hope adds clarity.	Page 20: “A greater number of bacteria were detected in paired specimens collected from children in the BioFire group when tested using the BioFire (n=23) compared to standard of care testing (n=11). However, we did not find this added knowledge to have significant clinical implications.”
Conclusion: I agree with the modest conclusions drawn and would stipulate that this was a small, single-center study and suggest that larger, multi-center studies are needed.	We have added this suggestion	Page 22: “However, as this was a small, single-center study, larger, multi-center studies are needed. Moreover, in the context of certain infections such as STEC, the reduced turnaround time has the potential to significantly alter clinical care.”
Reviewer #2		
Line 59: Should also include rectal swab specimens here in addition to stool.	This sentence has been revised as suggested.	Page 4: “Participants had stool (and rectal swabs if stool was not immediately available) tested using the routine microbiologic approach or to use of a device (BioFire FilmArray Gastrointestinal Panel) which identifies 22 pathogens with a 1-hour instrument turnaround time.”
Line 141: Indicate here that patients were randomized to have stool and/or rectal swab testing performed using the BioFire panel or conventional testing. The information regarding home stool collection could also be moved here (lines 193-196).	We have revised this section as suggested.	Page 8: “Participants were randomized to have stool testing performed using the BioFire FilmArray Gastrointestinal Panel or routine testing for enteric bacteria. Rectal swabs were performed in the ED if stool specimens were unavailable. A stool specimen was requested of all children who

Comment	Response	Changes/Location
		provided a rectal swab initially as per routine specimen collection procedures. Home stool collection was performed by those unable to provide a sample while in the ED.”
Line 156: Further define what is meant by "known to be STEC positive". Does this mean a patient with a recent previous visit in which STEC was identified? Please clarify.	We have clarified this exclusion criteria.	Page 8/9: “5) had documented evidence of an STEC infection based on a specimen submitted for testing during the current illness;”
Line 179: Should say "rectal swab testing was performed" or "a rectal swab was obtained".	Revised as suggested.	Page 10: “If stool was unavailable, rectal swab testing was performed.”
Line 205: Further details of the assays or modalities for virus and parasite testing are needed.	Details have been added.	Page 11: Testing for viral, parasites, and Clostridioides difficile was also performed if requested by the responsible physician. Conventional virus testing was performed using an in-house, real-time, quantitative polymerase chain reaction virus panel that detects generic adenovirus, astrovirus, norovirus genogroups I and II, rotavirus, and sapovirus. ¹⁶ Conventional parasite screening was performed using the BDmax enteric parasite panel for Giardia , Cryptosporidium , and Entamoeba histolytica . Microscopy was performed if requested and additional history was provided to the laboratory. ¹⁷ C. difficile testing was performed using the Liaison C. difficile glutamate dehydrogenase (GDH) enzyme immunoassay (EIA; DiaSorin S.p.A.; Italy) to screen specimens. If the GDH EIA was positive the specimen was tested by polymerase chain reaction for the presence of

Comment	Response	Changes/Location
		toxin B on the Xpert® C. difficile test (Cepheid; Sunnyvale, CA). Invalid specimens on the Xpert® C. difficile were retested using the C. diff Quik Chek Complete® EIA (TECHLAB, Inc; Blacksburg, VA)].”
From Table 2 it appears that six patients had C. difficile toxin testing performed as standard of care but details of that assay do not appear in the Methods.	We have added details of conventional C. difficile testing.	Page 11: “ C. difficile testing was performed using the Liaison C. difficile glutamate dehydrogenase (GDH) enzyme immunoassay (EIA; DiaSorin S.p.A.; Italy) to screen specimens. If the GDH EIA was positive the specimen was tested by polymerase chain reaction for the presence of toxin B on the Xpert® C. difficile test (Cepheid; Sunnyvale, CA). Invalid specimens on the Xpert® C. difficile were retested using the C. diff Quik Chek Complete® EIA (TECHLAB, Inc; Blacksburg, VA)].”
Line 220: If rectal swabs were tested using the BioFire panel then details of test preparation with that specimen type should be included.	The process for preparation of the rectal swabs, which are eluted in 2 mL of modified Cary Blair media after collection, was the same as for stool that was collected in Cary Blair medium. After mixing 200 µL were taken for processing on the BioFire. We have added this to the text.	Page 12: “After mixing, a 200 µL aliquot of stool suspension in Cary-Blair transport medium was used for the BioFire FilmArray test, which was conducted according to the manufacturer’s instructions.”
Table 2: By ">1 positive" do the authors mean to indicate the number of patients in which more than one organism was detected?	Yes, that is the correct interpretation but to clarify we have relabeled the rows as: Organism detected ≥2 Organisms detected	
In the SOC testing, does shiga toxin	In the standard of care arm, prior to June	Page 10-11: “Prior to June 2019, all positive

Comment	Response	Changes/Location
positive refer to the PCR assay? It did not appear than an EIA was used for shiga toxin detection. Please clarify.	2019, all positive STEC's were based on culture with Shiga toxin detection performed using the Shiga Toxin Quik Chek (TechLab, Inc., Blacksburg, VA). After that time, it was based on PCR using the BDMax. We have added these details to the manuscript.	STEC's were based on culture with supplemental Shiga toxin detection performed using the Shiga Toxin Quik Chek (TechLab, Inc., Blacksburg, VA). After that time, it was based on PCR using the BDMax."
In the footnotes indicating children <2 years of age with a positive C. difficile toxin, the authors should add that the numbers were not included since this likely represents colonization.	We have revised the footnote to clarify this.	“§ 1 child < 2 years of age was positive for Clostridioides difficile based on detection of the toxin gene by polymerase chain reaction and is not included in the number reported as positive since this likely represents colonization. ‡5 children < 2 years of age were Clostridioides difficile toxin A/B positive on the BioFire test; these children are not included in the number reported as positive since this likely represents colonization.”
Additional details are needed for reference 2.	The reference has been updated.	

March 10, 2023

Dr. Stephen Freedman
University of Calgary
Pediatrics
28 Oki Drive NW
Calgary, Alberta T3B 6A8
Canada

Re: Spectrum00268-23R1 (Comparison of a Rapid Multiplex Gastrointestinal Panel with Standard Laboratory Testing in the Management of Children with Hematochezia in a Pediatric Emergency Department: Randomized Controlled Trial)

Dear Dr. Stephen Freedman:

Your manuscript has been accepted, and I am forwarding it to the ASM Journals Department for publication. You will be notified when your proofs are ready to be viewed.

Sincerely,

Paul Luethy
Editor, Microbiology Spectrum
